# Chitosan Oligosaccharide Attenuates Lipopolysaccharide-Induced Intestinal Barrier Dysfunction through Suppressing the Inflammatory Response and Oxidative Stress in Mice

**DOI:** 10.3390/antiox11071384

**Published:** 2022-07-17

**Authors:** Wenjing Tao, Geng Wang, Xun Pei, Wanjing Sun, Minqi Wang

**Affiliations:** 1Key Laboratory of Molecular Animal Nutrition, Ministry of Education, College of Animal Science, Zhejiang University, Hangzhou 310058, China; taowenjing1127@163.com (W.T.); wanggeng@zju.edu.cn (G.W.); pxpeixun@zju.edu.cn (X.P.); sunwanjing@zju.edu.cn (W.S.); 2Hubei Key Laboratory of Animal Embryo and Molecular Breeding, Institute of Animal Husbandry and Veterinary, Hubei Academy of Agricultural Sciences, Wuhan 430064, China

**Keywords:** chitosan oligosaccharide, lipopolysaccharide, intestinal barrier function, inflammation, oxidative stress, mice

## Abstract

This study was conducted to investigate the protective effect of chitosan oligosaccharide (COS) against lipopolysaccharide (LPS)-induced intestinal injury. The results demonstrated that COS improved the mucosal morphology of the jejunum and colon in LPS-challenged mice. COS alleviated the LPS-induced down-regulation of tight junction protein expressions and reduction of goblet cells number and mucin expression. The mRNA expressions of anti-microbial peptides secreted by the intestinal cells were also up-regulated by COS. Additionally, COS decreased pro-inflammatory cytokine production and neutrophil recruitment in the jejunum and colon of LPS-treated mice. COS ameliorated intestinal oxidative stress through up-regulating the mRNA expressions of nuclear factor E2-related factor 2 and downstream antioxidant enzymes genes. Correlation analysis indicated that the beneficial effects of COS on intestinal barrier function were associated with its anti-inflammatory activities and antioxidant capacity. Our study provides evidence for the application of COS to the prevention of intestinal barrier dysfunction caused by the stress of a LPS challenge.

## 1. Introduction

The intestinal mucosa not only serves as a selectively permeable membrane allowing nutrients to pass but also provides a dynamic physical barrier defending against the pathogens and toxins [1]. The integrity of intestinal barrier depends on the presence of a con tiguous layer of enterocytes and the intercellular junctions—in particular, tight junctions [2]. In addition, products secreted by the intestinal mucosal epithelia, including mucins and anti-microbial peptides, provide the first line of defense against pathogen invasion [3,4]. 

Impaired intestinal barrier function has been associated with a variety of gastrointestinal and systemic diseases, such as inflammatory bowel diseases (IBDs) coeliac disease, diabetes and arthritis; however, there have been no approved agents that target the epithelial barrier thus far [5]. Growing evidence suggests that excessive immune responses to pathogens, characterized by an increased release of pro-inflammatory cytokines, disturb the intestinal barrier function by increasing epithelial apoptosis and decreasing junctional protein expression [6,7]. Conversely, the increased intestinal permeability promotes the invasion of pathogens, thus, exacerbating the mucosal inflammatory response [8]. Moreover, pathogens can also induce the overproduction of oxidants, such as reactive oxygen species, leading to intestinal inflammation and barrier dysfunction [9]. Lipopolysaccharide (LPS), as a component of the cell wall of Gram-negative bacteria, have been reported to induce intestinal damage accompanied by the increase of pro-inflammatory mediators and oxidative stress in various animal models [10,11,12,13].

Chitosan is the deacetylated product of chitin, which is widely found in the exoskeletons of crab, shrimp and insects as well as the cell walls of fungi [14]. Chitosan oligosaccharide (COS), an oligomer of β-(1,4)-linked D-glucosamine, is prepared from the hydrolysis of chitosan [15]. Numerous studies have shown that COS possesses several biological properties, including anti-inflammatory, antioxidant, anti-microbial and immunoregulatory activities [16]. 

In vitro, COS has been shown to inhibit LPS-induced pro-inflammatory cytokine production in several cell models, including macrophages, microglia, endothelial cells and epithelial cells [17,18,19,20,21]. Additionally, COS has been found to restore the LPS-induced reduction of transepithelial electrical resistance in the epithelium [20,21]. 

In vivo, COS was demonstrated to alleviate systemic inflammatory and oxidative damage to the liver, kidneys and lungs in LPS-challenged mice [22]. Moreover, COS might ameliorate mucosal damage and colitis induced by dextran sulfate sodium (DSS) in mice and reduce intestinal inflammation in LPS-challenged piglets [21,23]). Liu et al. reported that COS supplementation could reduce the incidence of diarrhea and improve the intestinal morphology in weaned pigs challenged with *Escherichia coli* K88 [24]. However, until now, there was no study that demonstrated the effects of dietary COS supplementation on intestinal barrier function in LPS-challenged mice. Whether the anti-inflammatory and antioxidant activities of COS are beneficial for improvement of intestinal barrier function has not been clarified.

Therefore, in view of the foregoing, we hypothesized that COS could attenuate intestinal barrier dysfunction induced by LPS in mice, and these beneficial effects of COS may be associated with its anti-inflammatory and antioxidant actions. In the present study, the intestinal morphology and ultrastructure and the expression of tight junction proteins, mucin and anti-microbial peptides were investigated to reveal the effects of COS on intestinal barrier function. Additionally, indices and gene expressions related to inflammation and oxidative stress were detected, and their correlation with intestinal barrier function was analyzed to verify the above hypothesis.

## 2. Materials and Methods

### 2.1. Materials

COS was purchased from Zhongkerongxin Biotechnology Co., Ltd. (Suzhou, China). The degrees of polymerization were 2–7, and the percentages of these oligomers were 5.60%, 36.19%, 34.20%, 21.17%, 2.54% and 0.30%, respectively. LPS (*Escherichia coli* O55:B5, #L2880) was obtained from Sigma–Aldrich (Saint Louis, MO, USA). Antibodies against adenosine monophosphate-activated protein kinase α (AMPKα) (#5831) and phospho-AMPKα (p-AMPKα) (#2535) were purchased from Cell Signaling Technology (Beverly, MA, USA). Antibodies against mucin-2 (Muc2, #27675-1-AP), Zonula occludens 1 (ZO-1, #21773-1-AP), occludin (#13409-1-AP) and β-actin (#20536-1-AP) were bought from Proteintech Group, Inc. (Wuhan, China).

### 2.2. Animals and Experimental Design

A total of 48 five-week-old male C57BL/6 mice were purchased from Slaccas Experimental Animal Co., Ltd. (Shanghai, China) and housed in plastic cages under a controlled environment (20–24 °C, 40–60% relative humidity and 12/12 h light/dark cycle). After one week of acclimatization, the mice were randomly divided into four groups (*n* = 12 per group): (1) Control, (2) COS, (3) LPS and (4) COS + LPS. Mice were orally administered with either phosphate-buffered saline (PBS) or COS (400 mg/kg BW) for 30 days. 

The doses of COS in this study were selected based on our preliminary experimental result. The body weight was recorded daily. All mice had free access to feed and water throughout the experimental period. On day 30, mice in the LPS and COS + LPS groups were intraperitoneally injected with LPS at a dosage of 20 mg/kg BW according to a previous study [22], while mice in the other two groups were injected with an equivalent amount of PBS. 

After fasting for 18 h, all mice were weighed, and blood samples were collected through retro-orbital plexus into tubes. After being kept at room temperature for 2 h, the tubes were centrifuged at 4 °C (1000× *g*, 10 min) to obtain the serum. Subsequently, all mice were euthanized, and segments of the duodenum, jejunum, ileum and colon were collected. The experimental procedures used in this study were approved by the Animal Care and Use Committee of Zhejiang University (ZJU20190087).

### 2.3. Histological Examination and Immunohistochemistry

The middle of the intestinal segments (including the duodenum, jejunum, ileum and colon) was fixed in 4% (*w*/*v*) paraformaldehyde (pH 7.0) for 24 h and then embedded in paraffin. Sections (5 μm thickness) of each intestinal segment were stained with hematoxylin-eosin (H&E) or periodic acid-Schiff (PAS). Images were acquired with a DM3000 microscope (Leica Microsystems, Wetzlar, Germany). The villus height and crypt depth of the duodenum, jejunum and ileum were measured using Image-pro plus 6.0 software (Media Cybernetics, Inc., Rockville, MD, USA). The histological damage of the colon was scored as described previously [25]. PAS staining was used to detect the numbers of goblet cells in the jejunum and colon.

Immunohistochemistry staining of Muc2 in jejunum and colon was performed according to a previous method [26]. The sections were deparaffinized with xylene three times for 15 min each, rehydrated in a graded series of ethanol (100%, 95%, 90%, 80% and 70%) for 5 min at each step, submitted to antigen retrieval using Tris-EDTA buffer (pH 9.0) and then incubated with 3% hydrogen dioxide in the dark for 25 min. After blocking with 3% BSA for 30 min, the sections were incubated with primary antibody against Muc2 for 12 h at 4 °C and then bound with secondary antibody for 50 min. Images were obtained using a DM3000 microscope, and the integrated optical density (IOD)/area of the positive staining was analyzed using Image-pro plus 6.0 software.

### 2.4. Scanning Electron Microscopy (SEM) and Transmission Electron Microscopy (TEM)

The jejunum segments were fixed with 2.5% glutaraldehyde for 24 h and then incubated with osmium tetroxide (1%) for 1.5 h. Subsequently, they were dehydrated in a graded series of ethanol (30%, 50%, 70%, 80%, 90%, 95% and 100%) for 15 min at each step and then dried in a Hitachi Model HCP-2 critical point dryer (Hitachi, Tokyo, Japan). The specimens coated with gold-palladium were observed by Hitachi SU-8010 scanning electron microscope (Hitachi, Tokyo, Japan). 

The samples for TEM were prepared as described previously [27]. After being fixed and dehydrated by the procedure above, the jejunum segments were transferred into pure acetone for 20 min, placed in a mixture of pure acetone and Spurr resin mixture (1:1 for 1 h and 1:3 for 3 h) and then transferred into Spurr resin mixture for 24 h. After being heated at 70 °C for 9 h, the specimens were cut into ultrathin sections (80 nm) by a Leica EM UC7 Ultramicrotome (Leica, Germany). The sections were stained with uranyl acetate and alkaline lead citrate for 15 min and then visualized by Hitachi H-7650 transmission electron microscope (Hitachi, Tokyo, Japan). The microvillus height was measured using Image-pro plus 6.0 software.

### 2.5. Enzyme-Linked Immunosorbent Assay (ELISA) and Biochemical Analyses

The level of diamine oxidase (DAO) and D-lactate (D-LA) in the serum were determined using ELISA kits (MLBIO Biotechology, Shanghai, China) according to the manufacturer’s instructions. The jejunum and colon tissues were homogenized with cold saline (1:9, *w*/*v*) and then centrifuged at 12,000× *g* for 15 min at 4 °C. The supernatant was collected to detect the levels of tumor necrosis factor-α (TNF-α), interleukin-1β (IL-1β) and interleukin-6 (IL-6) using ELISA kits (MLBIO Biotechology, Shanghai, China). The activities of myeloperoxidase (MPO), total antioxidant capacity (T-AOC), glutathione peroxidase (GSH-Px), superoxide dismutase (SOD) and the contents of malonaldehyde (MDA) were measured using commercial assay kits (Nanjing Jiancheng Biomedical Company, Nanjing, China) according to the manufacturer’s instructions. The protein concentration in supernatants was used to normalize these indices except for the MPO activities.

### 2.6. Real-Time PCR

The total RNA in the jejunum and colon tissue was isolated using TRIzol reagent (Invitrogen, Waltham, MA, USA) and reverse-transcribed into cDNA using a PrimeScript RT reagent kit (TaKaRa, Dalian, China). Real-time PCR was performed on the CFX96 Real-Time PCR system (Bio-Rad, Hercules, CA, USA) using SYBR Green (TaKaRa, Dalian, China). The primer sequences were detailed in Table 1. The thermal cycle conditions were 95 °C for 30 s and 40 cycles of amplification (95 °C for 5 s and 61 °C for 35 s). The mRNA expressions of the target genes were normalized to that of β-actin and determined using the 2^−^^△△CT^ method.

### 2.7. Western Blotting

The jejunum was lysed with RIPA buffer (Beyotime Biotechnology, Shanghai, China) to extract the total protein, and the concentrations were detected using a BCA assay kit (KeyGen BioTech, Nanjing, China). Thirty micrograms of protein samples were separated with 9% SDS-PAGE and then transferred onto PVDF membranes (Millipore Corp., Bedford, MA, USA) for Western blot analysis as described previously [26]. The protein bands were visualized using a ChemiScope 6000 chemiluminescence imaging system (Clinx Science Instruments, Shanghai, China) with the enhanced chemiluminescence reagent (Beyotime Biotechnology, Shanghai, China) and were analyzed using Image J software (National Institute of Health, Bethesda, MD, USA).

### 2.8. Statistical Analysis

All data were analyzed by one-way ANOVA using SPSS 20.0 software (SPSS, Chicago, IL, USA). The data are presented as the means ± SE, and Duncan’s tests were conducted to examine the differences among treatments. Nonparametric Spearman’s correlation coefficients were analyzed using GraphPad Prism 7.0 (San Diego, CA, USA). Differences among means were considered significant at levels of *p* < 0.05.

## 3. Results

### 3.1. Effects of COS on Body Weight

As shown in Figure 1A, there were no significant differences in body weight between mice treated with and without COS during the 30 days of gavage trial, suggesting that COS supplementation at a dosage of 400 mg/kg BW exhibited no negative effects on the growth of mice. However, after injection with PBS or LPS, no obvious differences in body weight loss were observed between the control group and the COS group, and the body weight loss of the LPS group was higher than that of the control group, whereas pre-treatment with COS attenuated the body weight loss caused by LPS (Figure 1B).

### 3.2. Effects of COS on Intestinal Morphology and Ultrastructure

Representative H&E stained images of the small intestine and colon are shown in Figure 2A–D. In the three regions of the small intestine, no significant differences in the villus height, crypt depth and villus height/crypt depth ratio (V/C ratio) were observed between the mice in the control group and the COS group (Figure 2E–G). The mice in the LPS group demonstrated decreased villus height and V/C ratios in the duodenum, jejunum and ileum and increased crypt depth in the duodenum and jejunum compared to those in the control group, whereas the mice in the COS + LPS group exhibited increased villus height in the jejunum and ileum, decreased crypt depth and increased V/C ratios in the jejunum compared to those in the LPS group (Figure 2E–G). 

In addition, the epithelial surface and crypt in the colon were damaged by LPS, whereas pre-treatment with COS attenuated LPS-induced colonic damage, leading to a lowered histopathological score (Figure 2D,H). However, COS alone did not change the histopathological score in the colon as compared with the control group (Figure 2H).

Subsequently, the jejunal ultrastructure was evaluated by conducting SEM and TEM, and it was not changed by COS alone as compared with the control group (Figure 3A–C). The protective effect of COS against LPS-induced jejunal villus damage was further confirmed via SEM (Figure 3A, upper). Moreover, the quantity and height of jejunal microvillus were decreased by LPS, whereas pre-treatment with COS inhibited LPS-induced jejunal microvillus damage (Figure 3A, lower, B and C).

### 3.3. Effects of COS on Intestinal Barrier Functions

As presented in Figure 4A,B, there were no significant differences in the serum levels of DAO and D-LA between the control group and the COS group; however, COS administration reduced the serum levels of DAO and D-LA elevated by LPS. 

In addition, COS alone did not change the mRNA expressions of ZO-1 and Occludin in the jejunum (Figure 4C). Mice in the LPS group demonstrated significantly down-regulated jejunal mRNA expressions of ZO-1 and Occludin compared to the control group, whereas pre-treatment with COS up-regulated these genes expressions in the jejunum. Western blot analysis displayed that the reduced jejunal protein expressions of ZO-1 and Occludin induced by LPS were reversed by COS administration (Figure 4D,E), which is consistent with the changes in the mRNA expressions. The TEM images indicate that LPS destroyed the structure of intercellular junctions in the jejunum, which was prevented by COS administration (Figure 3B). The AMPKα phosphorylated levels in the jejunum were not affected by LPS; however, COS administration increased the jejunal AMPKα phosphorylated levels in mice injected with both PBS and LPS (Figure 4D,F).

With PAS straining, we found that COS significantly attenuated LPS-induced decreases in the jejunal and colonic numbers of goblet cells; however, COS alone did not affect this parameter in the jejunum and colon (Figure 5A,B,E). The immunohistochemical analysis showed that Muc2 expressions were significantly down-regulated by LPS in the jejunum and colon, whereas COS administration significantly up-regulated them in LPS-challenged mice but did not affect them in mice injected with PBS (Figure 5C,D,F). 

In addition, there were no significant differences in the jejunal and colonic mRNA expressions of trefoil factor family 3 (TFF3) between the control group and the COS group (Figure 5G). Mice challenged with LPS had decreased TFF3 mRNA expressions in the jejunum and colon compared with the control group, whereas pre-treatment with COS up-regulated the colonic TFF3 mRNA levels. As shown in Figure 5H, no obvious differences in the jejunal mRNA expressions of regenerating gene type III (Reg3β and Reg3γ) were observed between the control group and the COS group. Mice in the LPS group demonstrated up-regulated Reg3β and Reg3γ mRNA expressions in the jejunum compared to the control group, whereas these parameters were higher in the COS + LPS group compared with those in the LPS group.

### 3.4. Effects of COS on Intestinal Inflammation

As shown in Figure 6, no significant differences in the levels of inflammatory cytokines (TNF-α, IL-1β and IL-6) and activities of MPO were detected in the jejunum and colon of mice between the control group and the COS group. The jejunal and colonic levels of TNF-α, IL-1β and IL-6, and activities of MPO were significantly increased after LPS challenge, whereas pre-treatment with COS significantly reduced all these parameters in the jejunum and colon.

### 3.5. Effects of COS on Intestinal Oxidative Stress

To investigate the effects of COS on intestinal oxidative stress, we evaluated the activities of antioxidant enzymes in the jejunum and colon. Compared with the control group, COS alone did not change the activities of antioxidant enzymes (T-AOC, GSH-Px and SOD) in the jejunum and colon (Figure 7A–C). LPS markedly decreased the activities of T-AOC, GSH-Px and SOD in the jejunum and the activities of T-AOC and SOD in the colon, whereas pre-treatment with COS significantly increased the activities of SOD in the jejunum and the activities of T-AOC and SOD in the colon. 

Additionally, there were no significant differences in the jejunal and colonic MDA levels between the control group and the COS group (Figure 7D). Mice in the LPS group demonstrated significantly higher levels of MDA in the jejunum and colon compared to the control group, whereas administration with COS significantly decreased the jejunal and colonic levels of MDA in LPS-treated mice.

Subsequently, the genes expressions of nuclear factor E2-related factor 2 (Nrf2) and related antioxidant enzymes in the jejunum and colon were evaluated. As shown in Figure 8, compared with the control group, COS alone did not change the mRNA expression of Nrf2, NADPH quinone oxidoreductase 1 (NQO1), heme oxygenase-1 (HO-1) and glutathione S-transferase class alpha 1 (Gsta1) in the jejunum and colon. 

LPS markedly down-regulated the mRNA expressions of Nrf2, NQO1 and Gsta1 in the jejunum and the mRNA expressions of Nrf2 and NQO1 in the colon, whereas pre-treatment with COS significantly up-regulated the mRNA expressions of Nrf2, NQO1 and Gsta1 in the jejunum and the mRNA expressions of Nrf2 in the colon. 

Additionally, mice in the LPS group demonstrated up-regulated HO-1 mRNA expression in the jejunum compared to the control group, and no obvious differences in colonic HO-1 mRNA expression were observed between the control group and the LPS group, whereas the jejunal and colonic HO-1 mRNA expressions were higher in the COS + LPS group compared with those in the LPS group.

### 3.6. Correlation between Intestinal Damage Biomarkers and Indices Related to Inflammation and Oxidative Stress

To further explore the correlation between intestinal damage and inflammation or oxidative stress, Spearman’s correlation coefficients were calculated as shown in Figure 9. The serum levels of DAO and D-LA were positively correlated with jejunal and colonic inflammation and negatively correlated with the jejunal and colonic antioxidant capacity. In the jejunum, the V/C ratio, microvillus height, protein expressions of ZO-1, occludin and Muc2, goblet cells number and TFF3 mRNA expression were negatively correlated with inflammation and positively correlated with the antioxidant capacity. 

In the colon, the histopathological score showed positive correlations with inflammation and negative correlations with antioxidant capacity. Conversely, the goblet cells number, Muc2 protein expression and TFF3 mRNA expressions showed negative correlations with inflammation and positive correlations with antioxidant capacity.

## 4. Discussion

The maintenance of intestinal health and homeostasis depends on the intestinal barrier function [28]. In this study, for the first time, we demonstrated that COS could attenuate intestinal barrier damage, which may be involved in suppressing intestinal inflammation and increasing antioxidant capacity in LPS-challenged mice. Our results suggest that COS is a potential effective dietary supplement to prevent LPS-induced intestinal barrier dysfunction as well as related intestinal diseases.

Previous studies have demonstrated that COS inhibited the increases in body weight of obese mice; however, it did not affect the body weight gain of normal mice [29,30,31]. Consistent with these findings, we found that continuous supplementation with 400 mg/kg BW COS for 30 days did not affect the body weight gain of mice. As expected, the LPS injection resulted in a significant increase in body weight loss compared to the control group, which was in accordance with early reports [32,33]. Notably, we found that pre-treatment with COS effectively prevented the decrease in body weight caused by LPS.

In the present study, we explored the effects of COS on the morphology of different intestinal segments. COS inhibited the decrease in villus height induced by LPS in the jejunum and ileum. However, only in the jejunum, the crypt depth was decreased, and the V/C ratio was increased by COS, suggesting that COS could prevent jejunal mucosal morphology damage by LPS, which was further supported by the improvement of the jejunal ultrastructure. COS also reduced the colonic morphologic damage induced by LPS challenge, indicated by a higher histopathological score. Notably, COS could reach the colon to exert a protective action because it is a non-digestible oligomer [21]. Therefore, our further research focused on the effects of COS in the jejunum and colon.

DAO is an enzyme exclusively synthesized in the enterocyte, and D-LA is an end metabolite of intestinal bacteria. These are released into the blood circulation when the intestinal mucosa is damaged [34]. In this study, we demonstrated that COS decreased the serum levels of DAO and D-LA elevated by LPS, indicating that COS could reduce the intestinal permeability in LPS-challenged mice. The tight junction is an important determinant of the epithelial barrier integrity, and ZO-1 and occludin are crucial for the structure and functions of the tight junction [2]. Our result was -consistent with a previous finding that LPS increased the intestinal permeability by down-regulating the expressions of jejunal tight junction proteins in mice [10]. However, dietary COS supplementation increased the expressions of tight junction proteins in the jejunum of LPS-challenged mice. Consistently, the TEM images confirmed that COS could improve the structure of intercellular junctions in the jejunum, which were destroyed by LPS. Taken together, these results indicated that COS improved the intestinal physical barrier function in LPS-challenged mice. 

Additionally, we also found COS administration increased the phosphorylation of AMPK in the jejunum. This was consistent with previous studies showing that COS activated the AMPK pathway in intestinal epithelial cells [35]. As a key cellular energy sensor, AMPK was reported to improve intestinal barrier function through promoting intestinal epithelial differentiation and enhancing ZO-1 assembly in Caco-2 [36]. As proved, COS could effectively activate AMPK; however, whether the beneficial effects of COS on intestinal barrier function were mediated by AMPK signaling needs to be further evaluated in vitro.

Goblet cells are one of the main cell types of the intestinal mucosal epithelium. Muc2 is the major member of intestinal mucin family secreted by goblet cells and is involved in the formation of the intestinal mucus layer [37]. Previous studies have reported that Muc2 expression was lowered in IBDs, and that a lack of Muc2 led to spontaneous colitis [38]. Therefore, the increased amount of goblet cells and production of Muc2 in the jejunum and colon by COS administration would be beneficial to restoring LPS-induced intestinal damage. 

TFF3 is secreted by goblet cells and plays a crucial role in epithelial restitution [39]. It has been reported that TFF3-deficient mice were more susceptible to DSS-induced colitis, whereas the administration of recombinant TFF3 attenuated DSS-induced colitis [39,40]. The increased colonic TFF3 mRNA expressions by COS in LPS-challenged mice indicated that the epithelial repair capacity was improved. Reg3β and Reg3γ, produced by Paneth cells, have bactericidal activity against pathogens [41,42]. Our results indicated that COS increased the mRNA expressions of Reg3β and Reg3γ in the jejunum of LPS-challenged mice, and this might be associated with the regulation effect of COS on the immune response. Collectively, these data suggested that COS improved LPS-induced intestinal biochemical barrier dysfunction of the jejunum and colon. However, the functions of these three anti-microbial peptides have only been preliminarily elucidated, and their roles in the beneficial effects of COS on intestinal barrier function require further research.

Intestinal inflammation has been revealed to be a vital inducing factor to intestinal barrier disruption [43]. The increased release of pro-inflammatory cytokine was observed in patients with IBDs [44], leading to decreased intestinal tight junction protein expression and then increased intestinal permeability [45]. Previous studies have demonstrated that COS suppressed the LPS-induced inflammatory response both in IPEC-J2 and in the jejunum of piglets and alleviated DSS-induced colitis [20,21,23]. Consistent with these findings, the current study demonstrated that COS could decrease the pro-inflammatory cytokine levels and MPO activity in the jejunum and colon of LPS-challenged mice, thereby, indicating a reduction in macrophage and neutrophil recruitment. Additionally, the correlation analysis results showed that intestinal barrier function was negatively correlated with inflammation, suggesting that the beneficial effects of COS on intestinal barrier function were associated with its anti-inflammatory activity. 

The gut microbiota have been considered to be an important factor for regulating intestinal barrier function [46]. Specially, metabolic disorders, such as obesity and diabetes, are usually accompanied by gut microbiota dysbiosis and intestinal barrier dysfunction, thereby, resulting in the leakage of LPS, produced by the gut microbiota, into the blood causing systemic inflammation [47,48]. Previous studies have demonstrated that COS could reverse the dysbiosis of gut microbiota and chronic inflammation in obese or diabetic mice [29,49,50]. However, the roles of the gut microbiota in improving intestinal barrier function and suppressing intestinal inflammation by COS treatment in LPS-challenged mice are yet to be defined. 

Oxidative stress has been shown to play an important role in disrupting intestinal barrier function and inducing gastrointestinal diseases [9]. Accumulating studies have reported that LPS can result in intestinal oxidative stress both in vitro and in vivo [13,51,52,53]. Our present study found that LPS decreased the activities of antioxidant enzymes in the jejunum and colon of mice, whereas COS increased the intestinal antioxidant capacity of LPS-challenged mice by restoring the activities of antioxidant enzymes. We also found COS decreased the levels of MDA in the jejunum and colon of LPS-challenged mice, indicating that COS lowered the degree of lipid peroxidation to suppress intestinal oxidative stress. To our knowledge, this is the first time that the beneficial effects of COS on attenuating intestinal oxidative stress induced by LPS in mice have been reported. 

The transcription factor Nrf2 plays a crucial role in maintaining cellular redox homeostasis, and the level of endogenous oxidants can be decreased by endogenous antioxidants and antioxidant enzymes via activation of the Nrf2 signal pathway [54]. We demonstrated that COS up-regulated the mRNA expression of Nrf2 and its downstream genes in the jejunum and colon of LPS-challenged mice, indicating that COS improved the intestinal antioxidant capacity by activating the Nrf2 pathway. Additionally, the correlation analysis results showed that the intestinal barrier function was positively correlated with the antioxidant capacity, suggesting that the beneficial effects of COS on intestinal barrier function are related to its antioxidant capacity.

## 5. Conclusions

In conclusion, our study demonstrated for the first time that dietary COS supplementation ameliorated the intestinal barrier disruption in LPS-challenged mice. The mechanisms of action might be closely related to improving the intestinal morphology, tight junction protein expression, mucin and anti-microbial peptide expression. The anti-inflammatory and antioxidant activities of COS might play important roles in its beneficial effects on intestinal barrier function. Our results provide helpful information regarding the application of COS in preventing intestinal barrier dysfunction caused by stress from LPS challenge. In our future work, we hope to investigate the underlying mechanism of the protective effects of COS against intestinal injury.

## Figures and Tables

**Figure 1 antioxidants-11-01384-f001:**
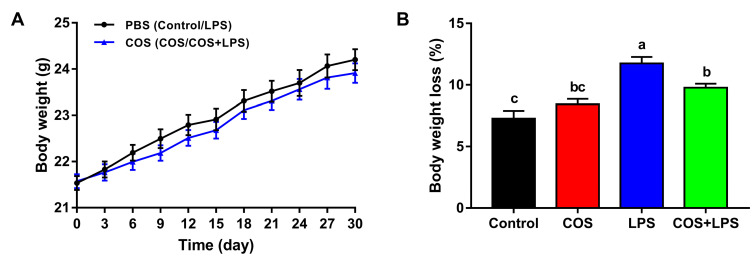
Effects of COS on body weight in mice. (**A**) Changes of body weight. (**B**) Body weight loss. The data are presented as the mean ± SE (*n* = 12). Values without a common letter are significantly different (*p* < 0.05). PBS, phosphate-buffered saline; COS, chitosan oligosaccharide; and LPS, Lipopolysaccharide.

**Figure 2 antioxidants-11-01384-f002:**
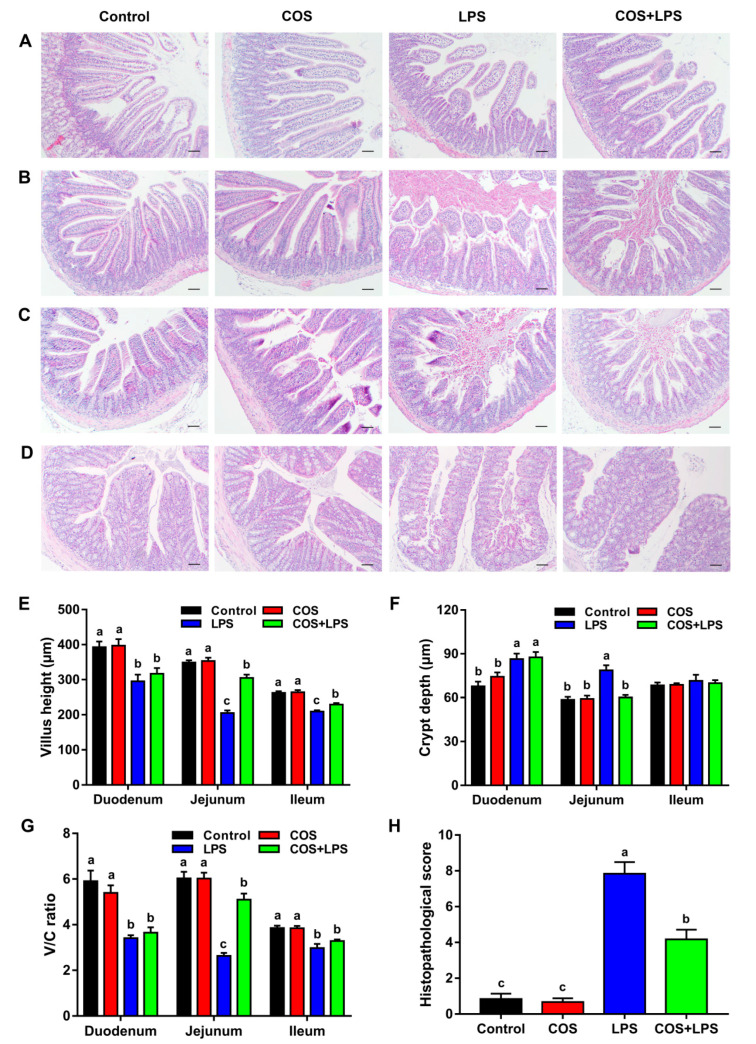
Effects of COS on the intestinal morphology in mice. Representative images of the duodenum (**A**), jejunum (**B**), ileum (**C**) and colon (**D**) stained with hematoxylin-eosin, scale bar = 30 μm. The villus height (**E**), crypt depth (**F**) and ratio of villus height to crypt depth (**G**) of the duodenum, jejunum and ileum. (**H**) The histopathological score of colon. The data are presented as the mean ± SE (*n* = 6). Values without a common letter are significantly different (*p* < 0.05). COS, chitosan oligosaccharide; LPS, Lipopolysaccharide; and V/C, villus height/crypt depth.

**Figure 3 antioxidants-11-01384-f003:**
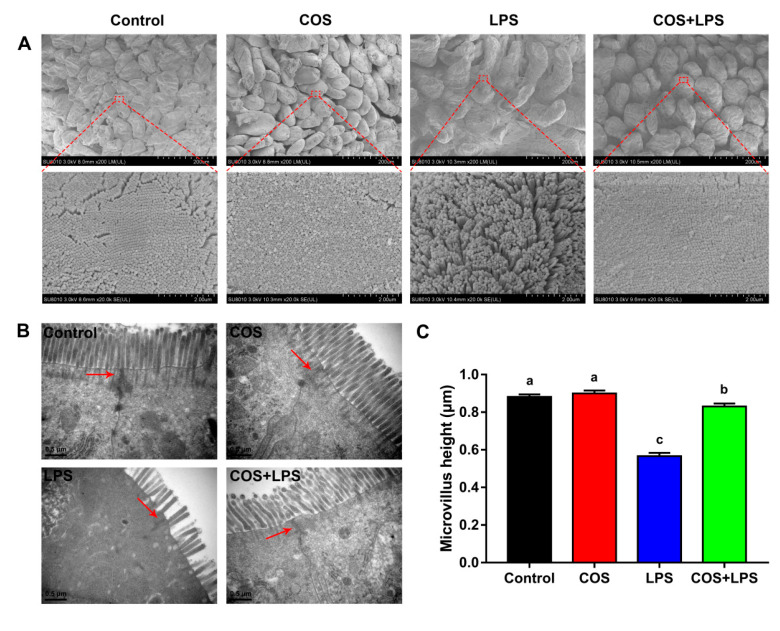
Effects of COS on the jejunal ultrastructure in mice. (**A**) SEM images (upper, 200×; lower, 20,000×). (**B**) TEM images (scale bar = 0.5 μm). (**C**) Microvillus height of jejunum. The data are presented as the mean ± SE (*n* = 6). Values without a common letter are significantly different (*p* < 0.05). COS, chitosan oligosaccharide; LPS, Lipopolysaccharide; SEM, scanning electron microscopy; and TEM, transmission electron microscopy.

**Figure 4 antioxidants-11-01384-f004:**
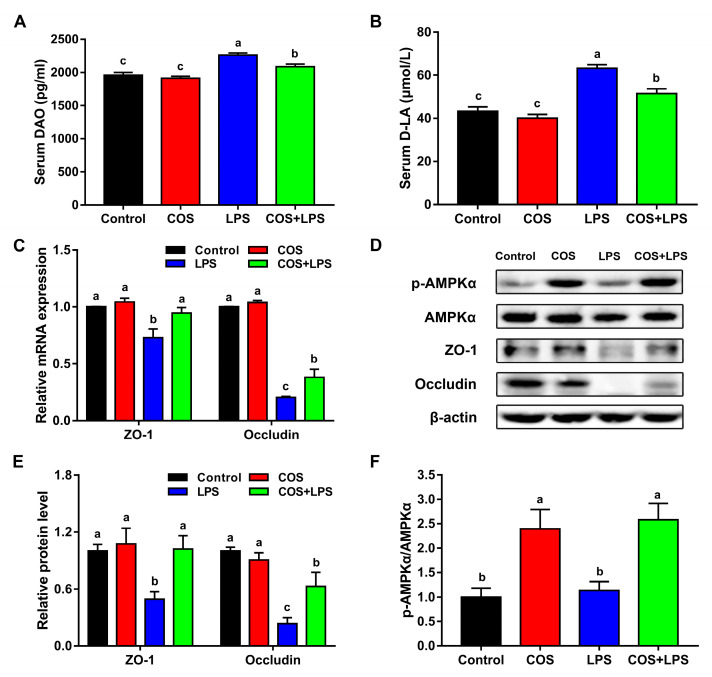
Effects of COS on intestinal permeability and intercellular junctions in mice. The level of DAO (**A**) and D-LA (**B**) in serum, *n* = 8. (**C**) The relative mRNA expression of ZO-1 and Occludin in the jejunum, *n* = 6. (**D**) Western blot analysis of ZO-1, Occludin, p-AMPK and AMPK protein expression in the jejunum. The density identification of ZO-1 and Occludin (**E**) and AMPK (**F**), *n* = 6. The data are presented as the mean ± SE. Values without a common letter are significantly different (*p* < 0.05). COS, chitosan oligosaccharide; LPS, Lipopolysaccharide; DAO, diamine oxidase; D-LA, D-lactate; ZO-1, Zonula occludens 1; and AMPKα, adenosine monophosphate-activated protein kinase α.

**Figure 5 antioxidants-11-01384-f005:**
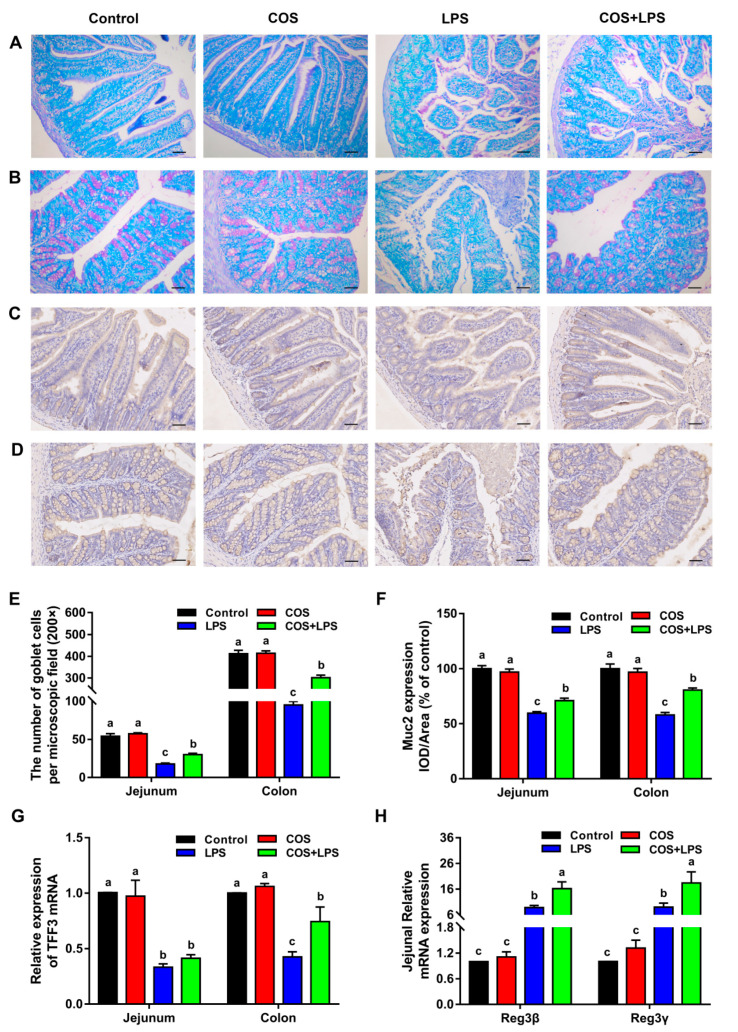
Effects of COS on biochemical barrier function of the jejunum and colon in mice. Representative images of the jejunum (**A**) and colon (**B**) stained with PAS, scale bar = 50 μm. Representative immunohistochemical staining of Muc2 in the jejunum (**C**) and colon (**D**). (**E**) The number of goblet cells per microscopic field. (**F**) The IOD/area of Muc2. (**G**) The relative mRNA expression of TFF3 in the jejunum and colon. (**H**) The relative mRNA expressions of Reg3β and Reg3γ in the jejunum. The data are presented as the mean ± SE (*n* = 6). Values without a common letter are significantly different (*p* < 0.05). COS, chitosan oligosaccharide; LPS, Lipopolysaccharide; Muc2, mucin-2; TFF3, trefoil factor family 3; and Reg3, regenerating gene type III.

**Figure 6 antioxidants-11-01384-f006:**
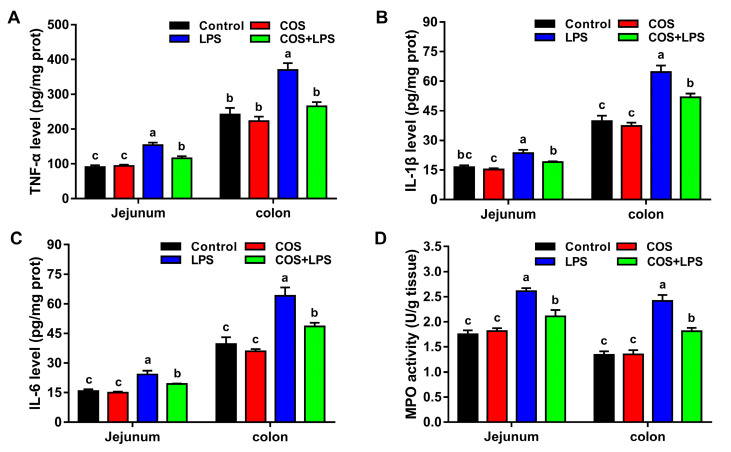
Effects of COS on inflammation in the jejunum and colon of mice. TNF-α (**A**), IL-1β (**B**), IL-6 (**C**) levels and MPO activity (**D**) in the jejunum and colon. The data are presented as the mean ± SE (*n* = 6–8). Values without a common letter are significantly different (*p* < 0.05). COS, chitosan oligosaccharide; LPS, Lipopolysaccharide; prot, protein; TNF-α, tumor necrosis factor-α; IL-1β, interleukin-1β; IL-6, interleukin-6; and MPO, myeloperoxidase.

**Figure 7 antioxidants-11-01384-f007:**
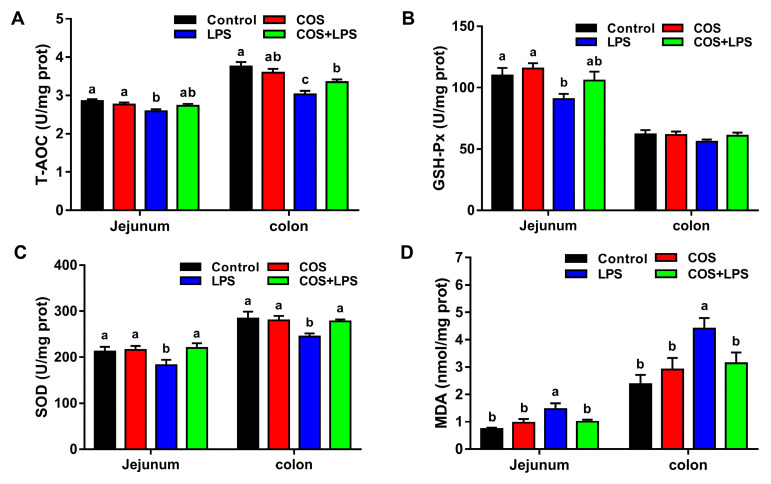
Effects of COS on antioxidant indicators in the jejunum and colon of mice. Activities of T-AOC (**A**), GSH-Px (**B**) and SOD (**C**) in the jejunum and colon. (**D**) Contents of MDA in the jejunum and colon. The data are presented as the mean ± SE (*n* = 8). Values without a common letter are significantly different (*p* < 0.05). COS, chitosan oligosaccharide; LPS, Lipopolysaccharide; prot, protein; T-AOC, total antioxidant capacity; GSH-Px, glutathione peroxidase; SOD, superoxide dismutase; and MDA, malonaldehyde.

**Figure 8 antioxidants-11-01384-f008:**
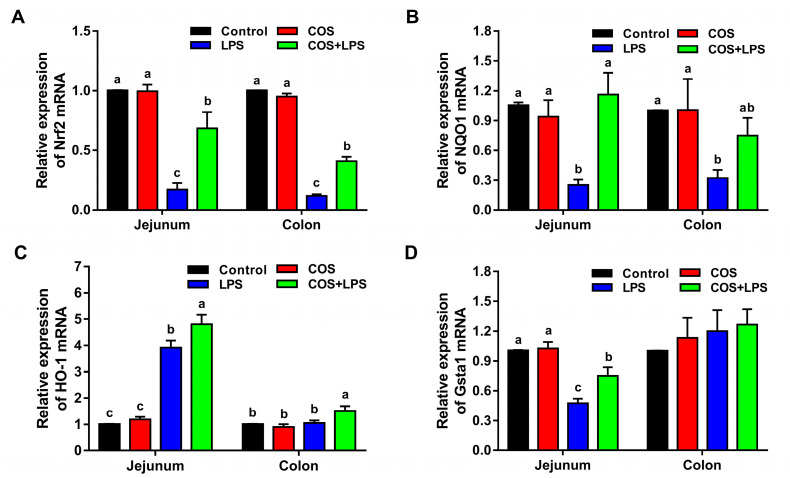
Effects of COS on mRNA expression of genes related to the antioxidant capacity in the jejunum and colon of mice. The relative mRNA expression of Nrf2 (**A**), NQO1 (**B**), HO-1 (**C**) and Gsta1 (**D**) in the jejunum and colon. The data are presented as the mean ± SE (*n* = 6). Values without a common letter are significantly different (*p* < 0.05). COS, chitosan oligosaccharide; LPS, Lipopolysaccharide; Nrf2, nuclear factor E2-related factor 2; NQO1, NADPH quinone oxidoreductase 1; HO-1, heme oxygenase-1; and Gsta1, glutathione S-transferase class alpha 1.

**Figure 9 antioxidants-11-01384-f009:**
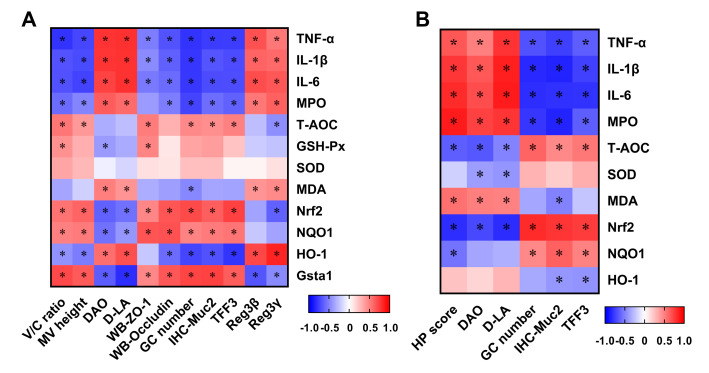
Correlations between intestinal damage biomarkers and indices related to inflammation and oxidative stress in the jejunum (**A**) and colon (**B**) of mice. Cells marked with an asterisk show significance following Spearman correlation for multiple comparisons (*p* < 0.05). V/C, villus height/crypt depth; MV, microvillus; DAO, diamine oxidase; D-LA, D-lactate; ZO-1, Zonula occludens 1; GC, goblet cells; Muc2, mucin-2; TFF3, trefoil factor family 3; Reg3, regenerating gene type III; TNF-α, tumor necrosis factor-α; IL-1β, interleukin-1β; IL-6, interleukin-6; MPO, myeloperoxidase; T-AOC, total antioxidant capacity; GSH-Px, glutathione peroxidase; SOD, superoxide dismutase; MDA, malonaldehyde; Nrf2, nuclear factor E2-related factor 2; NQO1, NADPH quinone oxidoreductase 1; HO-1, heme oxygenase-1; and Gsta1, glutathione S-transferase class alpha 1.

**Table 1 antioxidants-11-01384-t001:** Primer sequences used in Real-time PCR.

Target Genes	Accession Number	Forward Primer	Reverse Primer
ZO-1	D14340.1	AGGACACCAAAGCATGTGAG	GGCATTCCTGCTGGTTACA
Occludin	NM_008756.2	GGAGATTCCTCTGACCTTGAGTGT	TTCCTGCTTTCCCCTTCGT
TFF3	NM_011575.2	CCTGGTTGCTGGGTCCTCTG	GCCACGGTTGTTACACTGCTC
Reg3β	NM_011036.1	TGGGAATGGAGTAACAATG	GGCAACTTCACCTCACAT
Reg3γ	NM_011260.2	CCCGACACTGGGCTATGAAC	GGTACCACAGTGATTGCCTGA
Nrf2	NM_010902.5	CTACTCCCAGGTTGCCCACA	CGACTCATGGTCATCTACAAATGG
NQO1	NM_008706.5	AGGCTGCTGTAGAGGCTCTGAAG	GCTCAGGCGTCCTTCCTTATATGC
HO-1	NM_010442.2	GAGCAGAACCAGCCTGAACTA	GGTACAAGGAAGCCATCACCA
Gsta1	NM_008181.3	TGCCCAATCATTTCAGTCAG	CCAGAGCCATTCTCAACTA
β-actin	NM_007393.5	CGTTGACATCCGTAAAGACC	AACAGTCCGCCTAGAAGCAC

ZO-1, Zonula occludens 1; TFF3, trefoil factor family 3; Reg3, regenerating gene type III; Nrf2, nuclear factor E2-related factor 2; NQO1, NADPH quinone oxidoreductase 1; HO-1, heme oxygenase-1; and Gsta1, glutathione S-transferase class alpha 1.

## Data Availability

Data are contained within the article.

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
