# Peer review of "Chitosan Oligosaccharide Attenuates Lipopolysaccharide-Induced Intestinal Barrier Dysfunction through Suppressing the Inflammatory Response and Oxidative Stress in Mice"

_antioxidants, 2022, doi:10.3390/antiox11071384_

Round 1

Reviewer 1 Report

The manuscript by Wenjing Tao et al. investigate the protective effect of chitosan oligosaccharide (COS) in a murine model of lipopolysaccharides (LPS)-induced intestinal inflammation. 

The manuscript appears well presented and well written, a very large number of data are produced and analyzed providing evidence of protective effects of COS in intestinal inflammation. 

However, in my opinion, this study does not add relevant new information regarding what is already known on the anti-inflammatory effects of COS in gut. Indeed, previous works in literature, included some in the references list of the manuscript, already considered anti-inflammatory effects of COS in the same or in similar experimental models to that used in the reviewed manuscript.

In particular, 

1.     Huang B. et al (2016), using a piglet model of intestinal inflammation (intraperitoneal injection of LPS), provided evidence, by histopathological and biochemical analysis (inflammatory cytokine levels and NF-κB regulation), that dietary supplementation with COS significantly alleviated intestinal injury;

2.     Qiao Y. et al (2011), using the same model used in the reviewed manuscript (a murin model of intestinal inflammation induced by intraperitoneal LPS injection), showed the ability of dietary COS to reduce the serum levels of several proinflammatory markers and neutrophil infiltration. In addition, the authors showed, similarly to what is reported in the reviewed manuscript, that the observed protection was associated with antioxidant protection (increase of the levels of antioxidant enzymes and decrease in prooxidant markers) 

3.     Finally, Yousef M. et al (2012), using a mouse model of acute or chronic colitis induced by rectal instillation of 5% or 2.5 % respectively of dextran sulfate sodium (DSS), showed protective effects of dietary COS supplementation (reduced NF-kB activation and levels of proinflammatory soluble markers). Moreover, the authors showed that exposure to COS of intestinal epithelial cells prevents loss of epithelial barrier integrity induced by LPS or TNF-stimulation.

This considered, despite the high quality of the presented data, in my opinion the manuscript cannot be accepted in the present form for publication in Antioxidants.  It is very important that the authors clearly underline, by comparing their results with those present in the literature, the contribution of their study to the growth of knowledge on the anti-inflammatory action at the intestinal level of dietary COS supplementation. 

Reviewer 2 Report

I am glad to have an opportunity for reviewing this manuscript. This manuscript verified the anti-inflammatory and –oxidative effects of chitosan oligosaccharides in the LPS-induced mice model. The IBD and related disorders are rampant nowadays; therefore, I think that your results are helpful to find new therapeutic agents. However, I have strong doubts about your animal model. I hope that my opinions could help improve the quality of your study, and my comments on your manuscript are as follows:

I checked the whole manuscript, and I think that the flow is good and the paper is well-written. And your results and discussions are appropriate also.

However, As I said, I have strong doubts about your animal model.

1. At first, I found extremely few cases that used the LPS-induced mice model for testing the inflammatory response of the intestine. Why did the authors adopt this animal model and what are your references?

2. LPS is a well-known toxin that can cause inflammation and we also know it can cause breakage of the intestinal barrier system and loosen tight junctions. But LPS is a non-specific toxin; therefore, its administration causes sweeping inflammatory responses in the whole body. It means that extensive responses in the whole body can occur and I think that it does not match your target mechanism because the authors focused on topical regions.

3. Moreover, I think that this animal model is too aggressive to animals (can cause extremely strong inflammations in the whole body and it can result in death also. I think that this animal model cannot mimic human cases because it is deviant. 

Reviewer 3 Report

The manuscript by Tao et al., submitted for publication is an in vivo study investigating the effects of Chitosan Oligossacharide (COS), on the Lipopolysaccharides induced intestinal injury. This is an interesting work with potential clinical implications especially in the light of the microbiome modulation and potential chronic disease association.

The study is well designed and the manuscript is well organized and well written and fairly easy for the reader to follow. The reviewer would like to offer the following points for consideration by the authors towards the improvement of the manuscript:

1. How was the number of animals determined? Was there a power calculation?

2. How were the doses of COS and LPS determined?

3. It may be useful to include in the discussion section a short paragraph discussing aspects of the gut microbiome. One paper that might be found interesting is the following paper: Sikalidis, A.K.; Maykish, A. The Gut Microbiome and Type 2 Diabetes Mellitus: Discussing A Complex Relationship. Biomedicines 20208, 8. https://doi.org/10.3390/biomedicines8010008.

Round 2

Reviewer 1 Report

The authors have improved their manuscript. So, in my opinion, the manuscript can be accepted for publication.

Reviewer 2 Report

I am glad to have an opportunity for reviewing this manuscript. This manuscript verified the anti-inflammatory and –oxidative effects of chitosan oligosaccharides in the LPS-induced mice model. The IBD and related disorders are rampant nowadays; therefore, I think that your results are helpful to find new therapeutic agents. I hope that my opinions could help improve the quality of your study, and my comments on your manuscript are as follows:

1. In abstract section line 20, the authors need to eliminate “(nrf2)” because it is not repeated again in the abstract.

2. In the last sentence of the abstract section, you need to put some terminology that connotes “inflammation” because the authors verified inflammatory-induced intestinal barrier dysfunction.

3. Line 52: Please use italics for “in vitro”. And for line 55, use italics for “in vivo”.

4. Line 59: Liu et al. [24]

5. Line 61: Use italics for E. Coli

6. In materials, the authors should introduce detailed information about COS. Because COS is a mixture of chitosan oligomers. You presented the degree of polymerization of them but we cannot know the exact ratio of the dimer, tetramer, etc. Can you present more detailed information? I think that you can add the product number provided by the manufacturer.

7. How much volume of PBS with/without LPS did you inject into mice on day 30? I ask a question because I cannot understand the result of Fig.1B. Does the 'body weight loss' mean the ratio between before injection of PBS with/without LPS and just before sacrifice? Then, did the body weight decrease after the fast? In this case, the authors should write “we fasted mice for 18 h after the injection” in the methods.

And one more thing, in my experience, the 18 h fasting did not cause 7-8% weight loss in mice. Why the ratio of weight loss of mice was too high?

8. Lin 148: ‘reverse’

9. For Figure 6, does “mg prot” mean mg protein? Then, how did you determine protein amounts? I cannot understand how can you calculate interleukin levels into pg/mg protein unit. It is so strange because I do not think that you isolated proteins from samples before the assay. Or did you determine protein contents in supernatants firstly (using additional experiments like Lowry, Bradford…) and used it for calculation? In general, the unit of IL-1b level will be pg/mL when you monitored serum or will be the ng or pg/g tissue when you measured tissue samples.

And I cannot find the mouse interleukin ELISA kits for IL-1b and IL-6 on the site of MLBIO Biotechnology (The authors mentioned that you obtained those ELISA kits from this company).

10. In Figure 7, again I cannot understand how can you calculate enzyme activities and MDA levels into pg/mg protein unit.

11. I am really curious about the association between the expression of AMPKa and other genes. As you mentioned in the manuscript, AMPKa is a key regulator of energy metabolism and also it is a well-known key modulator for multiple signaling. That is, there are many papers that revealed the activation of AMPKa can upregulate anti-inflammation and -oxidation signaling. In Figure 4, the COS group showed increased phosphorylation of AMPKa 2.5 times compared to CON. This level is a quite high gap for AMPKa expression compared to other papers. However, all other factors throughout the whole results did not show different levels between CON and COS. I cannot really believe this phenomenon. Please recheck your results and if your results are correct, please explain why activation of AMPKa did not affect other mechanisms at all dissimilar to many papers.

Reviewer 3 Report

The authors have made a reasonable effort to address reviwer's points although the discussion on the microbiome would be more comprehensive and meaningful if the suggested paper was included. It would be important to note the connectioon to chronic diseases. I would thus suggest its inclusion.
